# Research Progress of Respiratory Disease and Idiopathic Pulmonary Fibrosis Based on Artificial Intelligence

**DOI:** 10.3390/diagnostics13030357

**Published:** 2023-01-18

**Authors:** Gerui Zhang, Lin Luo, Limin Zhang, Zhuo Liu

**Affiliations:** 1Department of Critical Care Unit, The First Affiliated Hospital of Dalian Medical University, 222, Zhongshan Road, Dalian 116011, China; 2Department of Critical Care Unit, The Second Hospital of Dalian Medical University, 467 Zhongshan Road, Shahekou District, Dalian 116023, China; 3Department of Respiratory, The First Affiliated Hospital of Dalian Medical University, 222, Zhongshan Road, Dalian 116011, China

**Keywords:** respiratory disease, interstitial pulmonary fibrosis, machine learning, artificial intelligence

## Abstract

Machine Learning (ML) is an algorithm based on big data, which learns patterns from the previously observed data through classifying, predicting, and optimizing to accomplish specific tasks. In recent years, there has been rapid development in the field of ML in medicine, including lung imaging analysis, intensive medical monitoring, mechanical ventilation, and there is need for intubation etiology prediction evaluation, pulmonary function evaluation and prediction, obstructive sleep apnea, such as biological information monitoring and so on. ML can have good performance and is a great potential tool, especially in the imaging diagnosis of interstitial lung disease. Idiopathic pulmonary fibrosis (IPF) is a major problem in the treatment of respiratory diseases, due to the abnormal proliferation of fibroblasts, leading to lung tissue destruction. The diagnosis mainly depends on the early detection of imaging and early treatment, which can effectively prolong the life of patients. If the computer can be used to assist the examination results related to the effects of fibrosis, a timely diagnosis of such diseases will be of great value to both doctors and patients. We also previously proposed a machine learning algorithm model that can play a good clinical guiding role in early imaging prediction of idiopathic pulmonary fibrosis. At present, AI and machine learning have great potential and ability to transform many aspects of respiratory medicine and are the focus and hotspot of research. AI needs to become an invisible, seamless, and impartial auxiliary tool to help patients and doctors make better decisions in an efficient, effective, and acceptable way. The purpose of this paper is to review the current application of machine learning in various aspects of respiratory diseases, with the hope to provide some help and guidance for clinicians when applying algorithm models.

## 1. Introduction

Machine learning (ML) is a subfield of artificial Intelligence (AI) and is based on a big-data-based algorithm that classifies, predicts, and optimizes according to previously observed data, using data to identify trends and complete specified tasks. ML contains two types of learning: supervised learning and unsupervised learning, and the range of techniques has gradually developed from simple linear models for complex neural networks with a large number of parameters. Multiple layers of “neurons” make up artificial neural networks (ANNs), which are based on the human brain and continuously process input data until they reach the output layer. Deep learning (DL), also known as convolutional neural networks (CNNs), is a recently developed variant of ANN that outputs data in a hierarchical manner, with successive layers evolving in between, processing incoming data in a fashion that includes both abstract high-level qualities, such as distinct objects, and simple low-level features, such as linearity [1].

A paradigm change in artificial intelligence is present along with CNN. In the early stages of AI research, the aim was to incorporate supervised learning into rule-based “expert systems” that could classify chest radiograph images as “normal” or “abnormal”. CNNs can more quickly distinguish between data availability and accuracy from large training datasets, for example, as the extensive usage of picture archiving and communication systems and electronic health records (EHRs). As an auxiliary tool for clinicians, machine learning has developed rapidly in medicine, especially in the application of respiratory diseases. Pulmonary imaging analysis can help distinguish normal lung tissue from ground glass opacities and honeycomb-like lung tissue changes, and assist in the differentiation of benign and malignant pulmonary nodules. Machine learning can assist in assessing indications for mechanical ventilation and the timing of weaning. In chronic respiratory diseases, it can assist in the assessment of pulmonary function to predict prognosis and treatment effect. In terms of respiratory biological information monitoring, it can help to monitor the early diagnosis of obstructive sleep apnea syndrome and reduce the occurrence of complications.

Early lung imaging of idiopathic pulmonary fibrosis (IPF) lacks evident specificity; thus, the accuracy of the diagnosis depends on the appropriate high-precision radiological imaging technology and is also constrained by the experience and expertise of radiologists and doctors. IPF is a chronic progressive inflammatory disease caused by a variety of reasons, with diffuse pulmonary parenchyma, alveolar inflammation, and interstitial fibrosis as the basic pathological lesions. In particular, the main clinical diagnostic methods of IPF include pulmonary imaging examination, lung biopsy, and pulmonary function test [2,3,4]. If the computer can help with the findings of an examination for fibrosis, it will aid in the early detection of such diseases, which is very beneficial to both patients and medical professionals.

Clinically, the situation of chest X-ray alone is complex, and it is difficult to effectively diagnose fibrotic lesions. High-resolution CT (HRCT) is usually characterized by “nonspecific interstitial pneumonia”, which shows honeycomb-like or stretch bronchiectasis or bronchiectasis in the bilateral subpleural base. Peripheral opaque ground-glass changes were most prominent in small nodules in the lower lobes. However, these are atypical lesions, which need to be differentiated from a variety of clinical disease-related factors, to further exclude autoimmune or drug factors, and exclude diseases other than interstitial pneumonia. In the past, based on artificial intelligence, we aimed to improve the diagnostic efficiency of patients with pulmonary fibrosis in a noninvasive way, and constructed a prediction model for pulmonary fibrosis, which achieved good clinical guidance.

At present, AI and ML have great potential and ability to change many aspects of respiratory medicine, which are the focus and hotspot of research. This article reviews the current application of machine learning in various aspects of respiratory diseases, hoping to provide some help and guidance for clinicians when applying algorithm models.

## 2. Application of Artificial Intelligence in the Respiratory System

### 2.1. Imaging Analysis of Pulmonary Nodules

Imaging analysis plays an integral role in the diagnosis and treatment of pulmonary diseases. DL and CNN are mainly used in medical imaging and have achieved promising results in lung nodule detection, as well as excellent performance in segmentation and classification of pulmonary nodules [5]. According to Siegel et al. [6], the 5-year survival rate is exceedingly dismal, and 55% of lung cancer patients have distant metastases at the time of initial diagnosis. Therefore, accurate classification and diagnosis of pulmonary nodules are essential to reduce the morbidity and mortality of early lung cancer. During image processing, CNN segments the image and isolates the analyzed object from the surrounding environment for analysis to evaluate the nodule size as a predictor of benign or malignant tumor. The volume method evaluates the sensitivity of nodule growth by reproducing and 3D-analyzing the size detection of nodules, and it is now regarded as the best technique for determining nodule size and growth [7].

Since 1980, several attempts have been made to develop computer-aided detection (CAD) algorithms for nodule segmentation. In the early 2000s, CAD for the identification of lung nodules began. Nodule segmentation, feature extraction, and classification of lesions into nodules and non-nodules are all aspects of conventional methods such as support vector machine (SVM). However, the traditional algorithm has a complex process and relies more on manual input, which limits the performance of CAD system to a certain extent [8]. However, DL algorithms do not need to rely on complex human dominant factors and may eliminate the innate barriers in traditional CAD systems. In 2015, Hua et al. [9] first published the detection results of lung nodules with a sensitivity of 73% and a specificity of 80% using DL algorithm on CT. Subsequently, a number of studies have shown that CAD and DL are superior to the traditional CAD algorithm. The sensitivity of nodule detection reached 85.4% [10]. In a study in 2018, LUNA16 and Ali Tanchi databases developed for detecting pulmonary nodules had a sensitivity of 81.7% and 85.1%, respectively [11]. The goal of CAD is to have low false positives while having high sensitivity. According to certain research, the sensitivity of nodule detection can reach 95%, but there are also various false positive rates [11]. In order to categorize nodules in 2019, Teramoto et al. [12] employed conventional CT in conjunction with early and delayed phase PET/CT. The findings revealed that 94.4% of malignant nodules were correctly detected. When compared to CT images alone and CT images plus early PET images, the accuracy of CT plus two-phase PET images in detecting benign nodules was greater by 11.1% and 44.4%, respectively. In a study by Hwang et al. [13], an algorithm that was trained on a dataset of 54,221 normal and 35,613 abnormal chest radiographs was able to differentiate between normal and tumor, active tuberculosis, pneumothorax, and pneumonia. This proved the superiority of the DL algorithm. The median area under the curve (AUC) for image classification and lesion identification was 0.979 and 0.972, respectively. When compared to several thoracic radiologists, the proposed algorithm performed noticeably better.

The prospective prediction of lung malignancy has been extensively fused and validated by radiomics and DL. DL models have been used to stratify patients, based on the likelihood of local and distant recurrence, to automate the segmentation of organs at risk in lung cancer radiation and identify individuals who would benefit from molecular targeted therapy and immunotherapy. The DL algorithm improves the performance under the influence of radiomics, and it is significantly better than the prediction based on clinical stage alone in the prospective cohort prognostic stratification test. This could help identify patients at higher risk of lung malignancy who could benefit from intensive treatment and/or more frequent follow-up after treatment [14].

### 2.2. Application of Racial Intelligence in Respiratory Monitoring in Critical Care Medicine

Mechanical ventilation is an important area of intensive care units (ICUs). It is a lifesaving tool that provides respiratory support to patients with respiratory failure in the ICU, and it is the focus of research in ML. Inappropriate mechanical ventilation may worsen lung injury, prolong the duration of mechanical ventilation, increase the risk of infection, and increase mortality. By collecting clinical parameters and laboratory results of critically ill patients, ML helps clinicians to predict the necessity of intubation within 24 h of admission to critically ill patients [15].

Hagan et al. [16] developed a personalized clinical prediction tool that can predict the respiratory support alert one hour in advance, increase the precision of clinical judgment, and reduce the incidence of inappropriate mechanical ventilation. In the study of patients receiving mechanical ventilation in ICU, based on the observation of an MIMIC study [17], whether mechanical ventilation patients need prolonged mechanical ventilation and further tracheotomy were predicted, with AUC of 0.82 and 0.83, respectively. The model may improve the prognosis of mechanical ventilation patients by performing tracheotomy as early as possible.

Prediction of infectious etiology is also an important direction of ML research. Sepsis and septic shock in severe infection are also one of the major life-threatening problems in intensive care. In many stages of sepsis, including early detection, prognostic assessment, mortality prediction, and clinical management, machine learning methods can be applied. Every hour of delay increases patient mortality, making early care in sepsis crucial. Initial sepsis prediction systems relied heavily on empirical clinical decision rules (CDRS), often using vital signs collected at the bedside. In 2018, Nemati et al. [18] developed an artificial intelligence sepsis expert (AISE) based on more than 27,000 and 42,000 intensive care unit patients from two hospitals. By removing data from electronic medical records and time series of high-resolution vital indicators, EHR data were combined with high-resolution blood pressure and heart rate measurements. For 12-h, 8-h, 6-h, and 4-h prediction of sepsis, the AI sepsis expert achieved an AUC in the range of 0.83–0.85. It was 0.85 at 4 h and 0.83 at 12 h before sepsis met the diagnostic criteria. In 2020, Akram et al. [19] retrieved five physiological markers from bedside monitors every minute in order to forecast the incidence of sepsis using a minimum collection of real-time physiological data. SVM classifiers categorize these data streams, which comprise heart rate, respiration rate, and blood pressure (systolic, diastolic, and mean blood pressure). With an average detection accuracy of 83.0% and an AUROC of 0.781, the model was able to predict the incidence of sepsis with high accuracy.

Since 2020, COVID-19 has spread around the world. Infected patients have developed severe respiratory symptoms, and may develop a variety of complications such as severe acute respiratory syndrome, sepsis, septic shock, and multiple organ failure. Effective methods to save costs and time are needed to reduce the burden of disease. In the search for potential treatments for COVID-19 among all available drugs, a study combined systems biology and artificial intelligence-based approaches. By using the GUILDify v2.0 Web server as an alternative approach, the effects of pirfenidone and melatonin on SARS-CoV-2 infection were confirmed. It also predicts the potential therapeutic effect of combination drugs on respiratory-related diseases [20].

Other applications of ML in the field of critical care medicine are also developing rapidly [21]. The incidence of postoperative pulmonary complications is high. In a study on high-risk chest patients, two machine learning models were produced through the identification, analysis, and fusion of respiratory failure risk factors. While the second model’s high sensitivity made it acceptable for clinical decision-making, the first model’s high accuracy and specificity made it suitable for performance evaluation [22]. In another study predicting pulmonary complications after gastrointestinal surgery [23], the ML algorithm was applied; the logistic regression model showed an AUC of 0.808, and the Gbm model showed an AUC of 0.814, which provided targeted support for clinical treatment.

There are also some machine learning models based on ready-made clinical data that can accurately and quickly identify ARDS phenotypes at the bedside [24]. For patients with initial blood parameters to further distinguish venous embolism from elevated D-dimer, machine learning models can improve the prediction rate of acute pulmonary embolism, help to narrow the indications of enhanced CT [25], and gain more rescue time.

### 2.3. Artificial Intelligence in Lung Function Assessment and Prediction of Chronic Respiratory Diseases

Chronic airway diseases mainly refer to asthma and chronic obstructive pulmonary disease, and the incidence and economic burden of developing countries are among the highest in the world [26]. It is characterized by ongoing inflammation, airway remodeling, obstruction, and recurrence, which significantly lowers quality of life and raises the possibility of hospitalization and death. For accurate prevention and individualized therapy, asthma is a diverse illness with numerous phenotypes and genotypes that must be appropriately differentiated. In recent years, various ML algorithms have employed genetic data in conjunction with clinical information, such as laboratory test results, to identify asthma phenotypes [27]. Spirometry and bronchial provocation tests, as well as eosinophil count analysis and fractional exhaled nitric oxide measurement, are employed in clinical practice to evaluate airflow restriction and hyperresponsiveness, allowing the identification of certain asthma phenotypes [28]. To identify asthma phenotypes realistically and precisely, additional research is still required.

A range of AI and ML methodologies have been applied to create affordable, secure, and efficient ways for the diagnosis of COPD. For instance, an “expert system” was built, employing the stages of questionnaire creation, network code development, pilot verification by expert panels, and clinical verification as an artificial intelligence diagnostic tool. The demographic, symptom, environment, and diagnostic test information was included in the questionnaire. This “expert system” obtained an overall accuracy of 97.5% in 241 patients during the clinical validation phase [29]. Subsequently, similar studies were conducted to construct artificial-intelligence-based software for the diagnosis of COPD from the clinical information of 1430 subjects [30]. According to the study, the developed software’s accuracy in diagnosing 50 COPD patients could reach 82%, which was much better than pulmonologists’ diagnostic performance (44.6 ± 8.7%). As a result, AI methods can significantly help clinicians when deciding how to diagnose COPD patients. ML was also used to mine and analyze transcriptional data extracted from human bronchial epithelial cells, reducing the reliance on lung function testing for early diagnosis of COPD. This led to the identification of aberrant expression of 15 genes in the disease, 10 of which were not previously reported as COPD biomarkers. In a recent study, five ML classifiers were used to separate healthy participants and COPD patients using 39 breath sound parameters, three lung function features, and data from 30 COPD patients and 25 control subjects. Diagnostic sensitivity, specificity, and accuracy of support vector machine and logistic regression were about 100% [31].

To investigate patterns related to respiratory outcomes in data gathered by remote monitoring devices, machine learning (ML) offers a potent answer. The oxygen desaturation index (ODI) and apnea–hypoxia index (AHI) were precisely calculated by an ANN (Nikkonen et al. [32]), utilizing only fingertip pulse oxygen saturation data as input. Using the manual event score as the gold standard, the median absolute error was 0.78 events/hour for AHI and 0.68 events/hour for ODI. Musavi et al. [33] created a DNN to label different stages of sleep using publicly available accelerated electroencephalogram (EEG) datasets to 84% reachable levels. Based on waveform analysis, Gholami et al. [34] created a random forest machine learning model with positive and negative predictive values of more than 90% to identify cyclic asynchrony. In 2020, Ma et al. [35], using an SVM-based strategy, used these features to create a classification model. The resulting model was used to diagnose OSAS. The system enables the collection of physiological data from a smartphone, data processing in the cloud, and real-time delivery of diagnostic findings to the smartphone. The results showed that the preliminary evaluation of the algorithm using real patient data found that its sensitivity, accuracy, and specificity were, respectively, 87.6%, 90.2%, and 94.1%.

Before a healthcare practitioner makes the initial diagnosis of possible apnea occurrences, several ML algorithms have successfully decreased the cost of diagnosis while enabling storage of physiological data from an Internet-based sleep monitoring device. A rising number of studies have shown that ML algorithms can successfully detect OSAS based on actual clinical data and pre-existing data, and the monitoring system’s ability to perform real-time diagnosis and remote monitoring makes it simple to use.

Notably, the AI program will also be able to use the currently developed features in combination with clinical data, medical records, past medical history, and patient demographic data to explore and predict the future role of OSA monitoring, which could significantly reduce disability rates and healthcare costs.

## 3. Application of Artificial Intelligence in IPF

IPF is a chronic progressive destruction of the lung disease. The average survival time is less than 5 years, and the early clinical manifestations of patients are not obvious, mainly manifesting as shortness of breath after activity, dry cough, recurrent lower respiratory tract infection, no obvious specificity, in the middle and late stage of progressive dyspnea, irreversible respiratory failure, and eventually death. The pulmonary function of the patients is delayed, usually with restrictive ventilation dysfunction, especially the reduction of forced vital capacity, total lung capacity, functional residual capacity, and diffusion capacity of the lungs for carbon monoxide (DLCO) [36]. Imaging plays a crucial role in the diagnosis of IFP. Chest X-ray is usually used as the initial means of imaging diagnosis of interstitial lung disease, but in some underdeveloped areas, chest X-ray is an indispensable part of imaging evaluation when critical patients are examined at the bedside. As a more sensitive imaging technique, high-resolution computed tomography (HRCT) is considered to be the core diagnostic tool for interstitial lung disease [37]. Abnormal lesions, such as irregular linear shadows, honeycombing, and reticular changes, observed on continuous HRCT can help radiologists and clinicians to identify specific interstitial lung disease lesions and assess the progression of the disease. However, at present, the assessment of the progression of interstitial lung disease mostly relies on the doctor’s visual analysis, which has certain subjective factors. Due to the limitation of clinical level, its accuracy and sensitivity are low. The detection rate of bronchoscopic lung biopsy for interstitial lung disease is also extremely low, and surgical lung biopsy is undoubtedly the gold standard for diagnosis. However, surgical lung biopsy has certain risks. For young patients with good lung function tolerance, it may have a certain guiding effect on their future treatment and prognosis. The risks of lung biopsy are so great that it may even increase the risk of death.

Deep learning approaches are used to identify, categorize, and segment ILD pictures on HRCT. At present, the most commonly used interstitial fibrosis mode CNN segmentation is U-Net [38]. Park et al. [39] analyzed 647 patients with lung segmentation by HRCT in ILD, and the accuracy reached 98%. Data augmentation is used in image processing to increase the amount of training data available. Common operations include image flipping, rotation, cropping, and scaling. Combined with data enhancement, the accuracy of fibrosis morphological classification can be improved to 78–91%. Using the 2011 ATS/ERS/JRS/ALAT criteria and the 2018 Fleischner Society criteria [36,40], Walsh et al. [41] developed a model to classify 1307 HRCT images of pulmonary fibrosis. Use of the UIP-HRCT model for pathological classification of fibrosis tissue can avoid the need for lung biopsy to a certain extent. Two studies [41,42] showed good performance in the diagnosis of IPF, which was close to expert level, and the diagnostic accuracy of the research tool was 78.9%, but there is also a high risk of bias due to the limited number of retrospective studies. Therefore, the value of current artificial intelligence technologies for the evaluation of ILD can only be reliably assessed by well-designed prospective controlled trials, and better evaluation algorithms and tools need to be further developed. Sikandar et al. [43] developed and trained the Forest model to evaluate 2424 subjects to predict the severity of pulmonary fibrosis patients, and the model achieved sensitivity and accuracy of 0.71 and 0.64, respectively. This model will help clinicians to diagnose IPF patients and assess the severity of the disease at an early stage, and make timely positive measures related to the treatment of IPF.

It has been demonstrated that certain CT-sensitive characteristics, such as reticulation and honeycomb, can accurately predict death in IPF patients. However, as prolonged progressive fibrotic ILD death is frequently unachievable End points, such as IPF, have the potential to be used as a substitute for therapy evaluation by changing the extent of the disease on high-resolution CT. In response, CALIPER has proven useful for tracking and forecasting disease [44,45,46]. Emphysema estimations based on threshold algorithms in a recent CALIPER-derived CT study [47] were considerably impacted by radiation dose, whereas the impact of dose reduction on texture-based algorithms was less thoroughly studied. There was a substantial association between CALIPER-derived CT characteristics and lung function results (FVC and FEV1%) in patients receiving treatment for pulmonary fibrosis, according to this study, which was the first to assess the impact of CT dosage adjustments on CALIPER performance. These results are in line with those of earlier retrospective investigations, but bigger prospective studies with longer follow-up times would undoubtedly be required to confirm the present results. Similar to PFT, CALIPER may be impacted by variations in lung volume; hence, it is crucial to have an experienced radiologist double-check the data. The use of conventional machine learning quantification software may be seen as a limitation given the increasing availability and complexity of deep learning algorithms; however, the datasets necessary to train such algorithms would require hundreds of thousands of cases, a difficult threshold to reach given the relatively low prevalence of this disease in the general population. When evaluating pulmonary fibrosis in IPF patients taking antifibrosis therapy, CALIPER parameters corresponded well with lung function, and CT dosage decrease had no effect on the software’s performance. When evaluating pulmonary fibrosis in treated patients, CALIPER can be a potent and objective addition to traditional lung function proxies in the absence of confounding factors affecting lung function.

We also proposed a prediction model for IPF [48]. By combining public datasets with clinical data, we were able to forecast X-ray models using the divide and conquer technique (Figure 1). We offered the upstream Attention-U-Net segmentation and downstream Inception-Res Net evaluation models, and their precision was in line with that of clinical specialists. In addition, the data processing speed was faster than that of clinical experts, which can significantly improve the diagnosis rate of diseases in areas where medicine is underdeveloped or there is a lack of experts. Based on the framework of upstream segmentation and downstream classification task, the complexity and final performance of the pulmonary fibrosis CT prediction model still need to be optimized (Figure 2). Our model classifies and segments the patient’s lung imaging images, and then quantifies the imaging images, which has certain generalization and application value, and objectively evaluates the degree of pulmonary fibrosis to achieve the best effect.

Although our model demonstrates predictive ability to a certain extent, machines can never completely replace humans. Previous studies also showed that neural networks have errors in distinguishing honeycomb-like changes from emphysema, and some imbalance training data can also lead to errors in imaging judgment of interstitial lung disease [39,49]. Deep learning requires training on a large number of samples, and training on larger datasets can help reduce misinterpretation of normal tissue as abnormal. Multicenter collaboration and publicly available image sets may help increase the data available for training. For further medical annotation, exhaustive and time-consuming input from experts is also required [50]. Although our model achieved good results in Doron training, it still needs to be further clinically validated with a large number of datasets to achieve objective and quantitative imaging evaluation of interstitial lung disease, and become a powerful tool for respiratory physicians and radiologists.

## 4. Challenges and Prospects

Since 2015, medical artificial intelligence research has been significantly accelerated, and respiratory medicine is a well-established profession. CNNs are becoming important resources for creating imaging biomarkers that can be used for diagnosis, prognosis, and treatment response prediction, and for the process of development and incorporation of machine learning models in the near future into regular clinical practice. The potential of CNNs beyond imaging in the field of lung function and physiological biosignal prediction remains enormous. However, a major limitation of this computational approach is the lack of a sufficiently large medical training dataset. Some application results still need to rely on clinical practice to confirm that medical institutions lack the ability to integrate information technology. To overcome these situations, large-scale digital information cooperation is needed, so that applications relying on large datasets can function more effectively with lower packet loss and higher packet retransmission rates.

ML can effectively and accurately improve the ability of diagnosis and treatment in pulmonary imaging analysis, pulmonary monitoring in critical care medicine, chronic respiratory diseases, and physiological and biological signals. The common goal is to improve the prognosis of patients and improve the quality of life. At present, interdisciplinary cooperation is a hotspot in the field of medical research, and the cooperative relationship between various medical sciences is crucial to the design of ML algorithms, because it will eventually bring new progress and changes to intelligent medical care [37,51]. We may see the potential for AI advice and assistance in the treatment of patients with complex diseases, such as those who have a number of ongoing health issues, or in the choice to have major surgery for critically ill, complex patients. Instead of whether computer algorithms can execute tasks better than people, how people will embrace and use new AI skills in the practice of medicine is the real concern. In the process of implementing ML, we need to conduct high-quality machine algorithm research on the premise of ensuring patient information security and efficacy, and explore the potential of ML in medicine to better work for the progress of the medical field. To assist patients and doctors in making better decisions in an effective, efficient, and acceptable manner, AI must develop into an undetectable, seamless, and impartial adjunct.

## 5. Conclusions

In conclusion, ML and AI have the potential to revolutionize many facets of respiratory care, particularly in the area of medical imaging, and they are already having a significant impact on the identification and classification of interstitial lung disorders. It is hoped that this brief review of artificial intelligence and machine learning in this article will be helpful to clinicians.

## Figures and Tables

**Figure 1 diagnostics-13-00357-f001:**
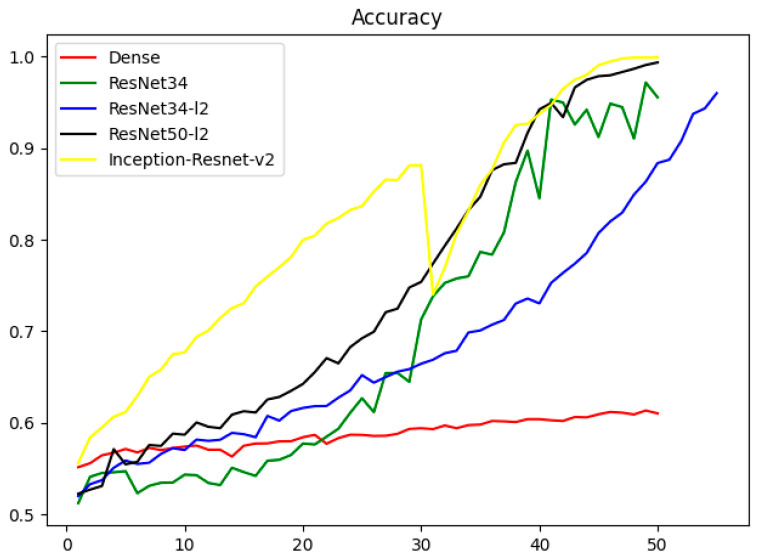
Comparison of the models in the X-ray classification task.

**Figure 2 diagnostics-13-00357-f002:**
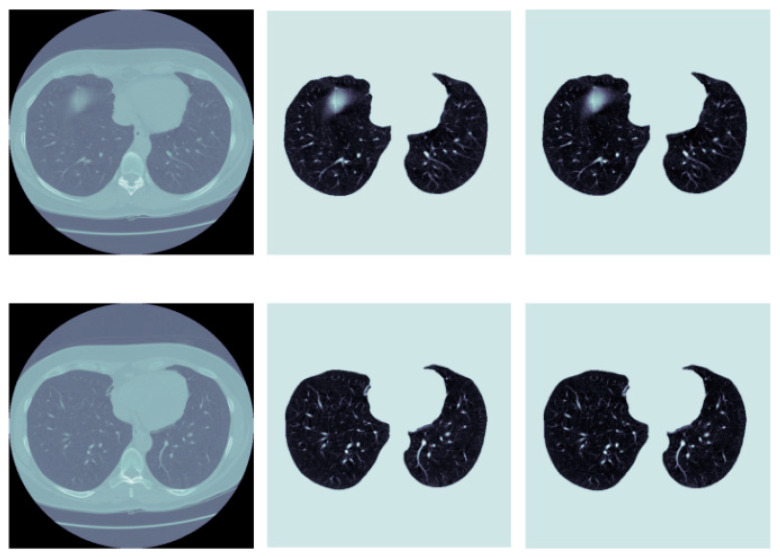
Comparison of CT segmentation effect.

## Data Availability

Not applicable.

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
