# Peer review of "Research Progress of Respiratory Disease and Idiopathic Pulmonary Fibrosis Based on Artificial Intelligence"

_diagnostics, 2023, doi:10.3390/diagnostics13030357_

Round 1

Reviewer 1 Report

Abstract need to be restructured

Introduction lacks contribution and structure of what the other section will provide.

Abstract should reflect the background knowledge on the problem addressed need to be added.

Abstract should reflect the wide range of applications and its possible solutions need to be added.

Abstract should reflect the problem addressed need to be justified with more details.

In Introduction section, the drawbacks of each conventional technique should be described clearly.

Introduction section can be extended to add the issues with respect to existing work

What is the motivation of the proposed work?

Literature review techniques have to be strengthened by including the issues in the current system and how the author proposes to overcome the same

Research gaps, objectives of the proposed work should be clearly justified.

Author Response

First of all, thank you very much for the reviewer’s suggestions,the manuscript has been corrected in accordance with the comments of the reviewer.

  • The abstract hadbeenrestructured, we added background knowledge on the problem and more details of the problem.
  • We addedthe propose of this paper.
  • We added contribution and structure ofthe other part of respiratory disease,and added a description of the drawbacks of conventional techniques, extend the introduction section and add existing work issues.
  • Research gaps were added as required and research objectives were further clarified.

Thanks again to the reviewers for their valuable comments.

Reviewer 2 Report

The authors present a review of the AI applied to respiratory diseases and IPF, the topic is interesting and the paper is well organized, however, it is necessary to clarify, in the title and the abstract, the type of research the review is considering.

There are writing errors in the document that must be corrected, for instance, the word "respiratory" is mistyped in the title.

The conclusion section could be ameliorated.

Although the presented papers considered in the review are interesting, the references should be updated, there are several interesting papers regarding AI applied to respiratory diseases in the past year. 

Author Response

First of all, thank you very much for the reviewer’s suggestions,the manuscript has been corrected in accordance with the comments of the reviewer.

1.The abstract was revised and some background knowledge was added to further clarify the purpose of the study and the type of review.

2.The manuscript was further proofread for language and spelling.

3.The conclusion section was further modified.

4.Citations were further updated as requested 

Thanks again to the reviewers for their valuable comments.

Round 2

Reviewer 1 Report

ACCEPTED